# The Theory of the Surface Wettability Angle in the Formation of an Oil Film in Internal Combustion Piston Engines

**DOI:** 10.3390/ma16114092

**Published:** 2023-05-31

**Authors:** Piotr Wróblewski

**Affiliations:** 1Faculty of Mechatronics, Armament and Aerospace, Military University of Technology, Sylwestra Kaliskiego 2, 00-908 Warsaw, Poland; piotr.wroblewski@wat.edu.pl or piotr.wroblewski@uth.edu.pl; 2Faculty of Engineering, University of Technology and Economics H. Chodkowska in Warsaw, Jutrzenki 135, 02-231 Warsaw, Poland

**Keywords:** friction losses, contact angle hysteresis, film thickness, critical shear stress, wettability

## Abstract

In internal combustion piston engines, the formation of an oil film is completely different from that seen in industrial machines. The molecular adhesion force at the interface between the surface coating of engine parts and the lubricating oil determines the load-carrying capacity and the ability to form a lubricated film. The geometry of the lubricating wedge between the surfaces of the piston rings and the cylinder wall is created by the thickness of the oil film and the height of the ring’s coverage with lubricating oil. This condition is affected by many of the parameters that characterize the engine’s operation and the physical and chemical parameters of the coatings used for the cooperating pairs. For lubricant particles that reach energies that are higher than the potential energy barrier regarding adhesive attraction at the interface, slippage occurs. Therefore, the value of the contact angle of the liquid on the surface of the coating depends on the value of the intermolecular force of attraction. According to the current author, there is a strong relationship between the contact angle and the lubrication effect. The paper shows that the surface potential energy barrier is a function of the contact angle and contact angle hysteresis (CAH). The innovation of the current work consists in examining the contact angle and CAH under the conditions of thin layers of lubricating oil, in cooperation with hydrophilic and hydrophobic coatings. The thickness of the lubricant film was measured under various speed and load conditions, using optical interferometry. The study shows that CAH is a better interfacial parameter for correlation with the effect of hydrodynamic lubrication. This paper presents the mathematical relationships relating to a piston engine, various coatings, and lubricants.

## 1. Introduction

In internal combustion engines, as a result of changes in the average working pressure found in the working chamber of the engine, changes in gas pressure occur in the labyrinth spaces of the ring seal. This, in turn, leads to varying pressures of the rings on the cylinder wall and changes in the thickness of the oil film. The thickness of the film is the result of many variables, including the inclination of the rings relative to the surface of the cylinder. The value of these changes depends on many of the geometrical parameters of the engine’s structural elements, the parameters of its operation, and the chemical parameters of the lubricating liquid. The properties of the surface of the material used, mainly its hardness, roughness, Young’s modulus, and surface energy, also have a significant impact on the distribution of the oil film. Many authors focus their attention on the design of the surface topography, ignoring the most important parameter, which is the hydrophilic and hydrophobic properties of the coatings. The impact of these parameters, as indicated by the author’s earlier works, is very important and their significance should not be overlooked [1]. Contact angle hysteresis (CAH) is one of the most important and classic elements of liquid droplet wetting in tribological systems. There are many papers on CAH determination [2,3,4]. According to the authors of [5,6,7,8], it has been recognized that significant contact angle measurements can be used to calculate constant surface tensions. In recent years, numerous methods have been used to measure the contact angle [9,10,11,12]. The most important surface parameters include self-cleaning, adhesion, low friction, wettability, hydrophobicity and/or hydrophilicity, and surface energy. These parameters largely determine the lubrication conditions in internal combustion piston engines, including the ability to create an oil film and plan for its thickness in the engine’s operating cycle. However, they are not fully taken into account in most theoretical research and simulation models. The parameters of the coatings used in internal combustion engines, such as surface energy, microgeometry, hydrophilicity, hydrophobicity, contact angle, and contact angle hysteresis, have been described in a small number of articles.

The results of research into the properties of coatings and liquids are included in previous studies [13,14,15]. Experimental comparisons were also carried out, but these used different scales and hydrodynamic conditions or nanoscale contacts [16,17]. Papers on macroscale contacts expressed more similar results [18,19,20,21,22,23]. An additional limitation of these results, when it comes to engineering models, is the fact that the experiments were sometimes performed in non-engineering conditions, i.e., using model surfaces or model liquids [19,20,21]. Studies using surfaces similar to those used in internal combustion engines and oils have been presented in [20,22,23]. Although the number of scientific works is constantly increasing in the field of wetting, there is still a lack of full understanding of the phenomena occurring in internal combustion engines. The lack of significant parameters in the models and the various theories correlating with each other cause a large discrepancy in the research results. This often applies when mapping the surface stereometry of cooperating elements, kinematics, and the dynamics of elements working in the main mechanism and engine timing. There are also difficulties when modeling the thermodynamic parameters of engines, in particular the effect of gas on changes in the position of the rings relative to the cylinder wall. This greatly affects the accuracy of the calculation of the minimum oil film thickness between the selected kinematic pairs of the engine. Currently, there are almost no scientific works presenting simulation models of the cooperation of pistons and piston rings with the cylinder surface, along with the introduction of wettability data, hydrophilic and hydrophobic properties, full surface stereometry, and the phenomena of the simultaneous flow of liquid and gas through the nanospaces of rubbing pairs. Research using an engine was conducted by the author for the first time and was described in detail in [1]. According to the author, determining and controlling the CAH is of great importance to the operation of these tribological systems.

Several experimental papers have been developed in the area of lubrication theory using CAH. Thanks to them, it is possible to learn some of the mechanisms of operation when applying this theory to internal combustion engines. Molecular dynamics (MDS) simulations were performed in [24]. It was found that the ultimate slip value increases with a greater wettability angle of the coating. The value of the slip length reaches up to thirty molecular diameters. This is the case for a contact angle of approximately 140°. Using the MDS for the water model, the wettability angle is the parameter that defines the slip control [25]. It was assumed in the research that the exponential characteristic defining the relationship between the slip length and the value of the wettability angle for liquid-repellent surfaces represents the rate of increase in the slip length. In another paper [26], measurements of the liquid penetration force between the sphere and the hydrophilic and hydrophobic coating were performed. The penetration substance was glycerol. The measurement was performed using a surface force apparatus (SFA). A slip occurred only in non-wet conditions. The particle image speedometer method can also be used to identify the occurrence of limit slips [27]. In this work, the apparent speed of sliding was noted when water flowed through a hydrophobic channel. An apparatus used for dynamic surface forces to measure the interaction forces in water was employed in [28]. The authors of this paper observed that water slippage increases with hydrophobicity. A study of slip and its length in a Newtonian fluid, using the SFA method, was performed in [29]. With regard to kinematic pairs lubricated hydrodynamically at low loads, studies were carried out in [18,30]. In these studies, glycerol and n-hexadecane lubricants were used. To a large extent, most studies on the interaction of the surface of a material with a liquid involve the measurement of drainage or hydrodynamic force. A comparison of the ability to form a lubricating film for different surfaces is presented in [31]. Interferometry was used to measure the micro scale with high accuracy. As part of these measurements, it was found that a surface characterized by high wettability and a small contact angle produces a greater thickness of lubricating film. This is related to the intermolecular attraction of the lubricating liquid to the surface layer of the coating. However, the results were inconclusive. In [32,33], different hydrodynamic drainage forces were obtained for the flow of water on a fully wetted tribological system. In [34], it was established that the coefficient of friction under EHL lubrication conditions does not depend on the hydrophobicity of the coating. This test was carried out on a PDMS target with a ball. Despite this, the results showed that in the boundary lubrication regime, the coefficient of friction decreased with the decrease in the contact angle. In [35], the hydrodynamic force was measured in various liquids with a surface covered with alkylsilane and an AFM colloidal probe. It was assumed that the slip length decreased along with the contact angle for non-polar liquids when the value of the contact angle was in the range of about 10° to 40°. For polar liquids with a contact angle of 60° to 100°, a slight relationship was found between the contact angle and the slip length. In these works, various conclusions were drawn regarding the influence of wettability. It should be noted that the lubricant flow in labyrinth seals or bearings is dynamic, while the contact angle is a static parameter reflecting the wettability range.

The aim and innovation of this work are to compare the effectiveness of the contact angle and CAH in correlation with hydrodynamic lubrication, using the example of a labyrinth seal employing the standard coatings and lubricating oils used in engines. According to the author, these results are only credible when comparing tribological and engine bench tests. The results of this work show the essence of the problem of hydrophilic and hydrophobic interactions and their effects on the decomposition of the oil film. With modern technologies for creating novel coatings and additives to lubricating oils, this may be the future of reducing friction losses in internal combustion engines.

## 2. Materials and Methods

In order to carry out the tests, an optical station was built for testing sliders with a constant inclination in accordance with that used in [36]. Sliding friction is achieved by means of a glass disk rotating on a stationary inclined slider. The angle of inclination can be freely adjusted depending on the desired results. The thickness of the oil film is measured by interferometry (Figure 1). The initial reflection of the incident beam at the liquid-solid interface is supported by a sputtered chromium coating on the glass surface. When the disk rotates, the hydrodynamic effect causes the slider to lift. In the test, the number of interference fringes is identified to determine the slope. When the test equipment is turned off, the slider falls freely to the disk surface. The exact determination of the film thickness for a given reference speed, in accordance with the shift of kinematic pairs in the internal combustion engine, is based on a change in the order of the fringes and the intensity of the selected point of the interferogram during the stopping period, employing a multi-beam intensity approach [37]. In a piston-powered internal combustion engine, the average thickness of the oil film under load conditions, e.g., the expansion stroke, does not exceed 10 μm; this maximum value was adopted in the experimental studies. The average measurement error for this rig was less than 8 nm.

In the experimental study, a glass disc covered with a layer of chromium and SiO_2_ was used to ensure the appropriate resolution conditions for the optics and surface protection. The reflectivity of the chrome coating is about 20%. In the test, 5 different coatings were sprayed on the slider of the device. The size of the slip plane was 5.15 mm by 10.20 mm. The material properties and their roughnesses are summarized in Table 1. Steel intended for the production of rings (slider 1) was used as the basic material in the test. This stainless steel is designated as 1.4112 (X90CrMoV18), with high hardness and abrasion resistance. Various multi-layer and single-layer nanocoatings were applied to this steel (sliders 2–5). Hydrophobic and hydrophilic coatings were adopted. The roughness of all samples was measured in nanometers. Three lubricants, 65 wt % glycerol, 99 wt % glycerol, and lubricating oil, were used as the base fluids. The properties of these greases are shown in Table 2. The contact angle (CA) and the hysteresis of the contact angle (CAH) were measured using a measuring device for selected liquids and coatings. Depending on the sample, a drop of liquid would need a short exposure time to stabilize the optical parameters of the drop. This lasted for a maximum of 10 s. The measurement was considered valid after stabilization. The test was carried out 5 times for the same liquid and a given sample. For this purpose, a measurement error of no more than 3.5% was obtained for all test samples and lubricants. Before testing, all samples were bathed in an ultrasonic cleaner in 99% alcohol. Any contamination was removed with a special cloth. The top layer of the sample was then dried for about 3 min.

Table 3 summarizes the results, grouped according to the different liquids and base materials. In ideal conditions, the unique contact angle of a liquid drop on a perfectly smooth and flat solid surface can be predicted, based on Young’s equations [38]. In experimental test conditions, the range of contact angles can be measured. The upper and lower limits of the range are established by increasing or decreasing the contact angles. The hysteresis of the contact angle (CAH) can be represented as the difference between the increasing and decreasing contact angles (CA). The experiment used the sedentary method to determine the CAH. A droplet of the test liquid with a volume of less than 10 μL was deposited on the surface layer of the steel or coated steel sample. Liquid was added to the drop until the line of contact shifted. At the appropriate moment, a measurement was made of the advancing contact angle. Alternatively, liquid was taken from the droplet. At the time of retraction, the contact angle was reached as the line of contact moved. The tests were carried out at an ambient temperature of 17.0 ± 0.5 °C and a humidity of 30 ± 1%. A new glycerol sample was used for each research group. It is well known that glycerol is hygroscopic. Therefore, a time change in the viscosity of 99 wt % was established. It was assumed that in the area of research, the viscosity would only drop by 1.1% in 20 min. During the test, the viscosity of all the liquid samples was constant. No greater changes in viscosity than 0.1% were recorded; therefore, it was assumed that the viscosity of the liquid samples was unchanged. Angle wettability and angle hysteresis studies were carried out using a goniometer (Model 790 High-Speed Automated Goniometer/Tensiometer with DROPimage Advanced (p/n 790-G1).

The given materials were adopted due to their having divergent contact angle properties with similar material properties. Some of these materials are described in previous publications [1,39,40].

## 3. Results and Discussion

In a piston-powered internal combustion engine, as part of the movement of the piston in the cylinder sleeve, variable instantaneous speed is seen in the moving kinematic pairs. This is due to the reciprocating movement of the piston. The ring pack moves stochastically in the piston grooves. Therefore, it is difficult to determine the mutual position of the moving planes of the piston and piston rings, relative to the cylinder surface. Nevertheless, it is possible to determine their mutual position with high probability, depending on the dynamic viscosity of the lubricant, the geometry of all elements of the main engine mechanism, the rotational speed of the crankshaft, and the thermodynamic conditions in the combustion chamber. The distribution of the thickness of the oil film between the piston rings and the cylinder wall is affected by a number of engine operating parameters [41]. On the basis of simulations and experimental data, it is possible to determine the approximate thickness of the oil film between the given engine components. The smallest film thickness occurs between the upper piston ring and the cylinder wall. On average, it ranges from 0.1 to 15 μm, depending on the engine stroke, the shaft speed, and other parameters characterizing the engine load conditions. As is well known, oil parameters play a very important role in this context and affect the hydrodynamic pressure values and the possibility of mixed friction or boundary friction conditions.

In the tests, measurement of the thickness of the lubricating film was performed. Different loads were assumed for the selected slider sets with applied coatings and lubricating fluid. The angle of inclination was constant during the test run. The name of the slider was derived from the name of the coating. During the interferometric test, the number of fringes at different speeds was read and analyzed. In the test, the same number of evenly arranged fringes shifted on the given interferograms with different speeds, which means that the slider inclination was unchanged at the time of the test and no elastic deformation of the contact surface of the kinematic pairs was observed. All tests were carried out under conditions of hydrodynamic lubrication, reflecting the conditions of fluid friction between engine components. Experiments with 65% by weight of glycerol were performed using various sliders, with a constant load of 5 N and a constant slope of 1:1875, as with the inclination of the plane ring relative to the cylinder face for symmetrical and asymmetric parabolic shapes. 

Figure 2 shows the variation of film thickness versus speed for the 65 wt % test of glycerol solution. In order to better understand the issues of hydrophilicity and hydrophobicity, two theoretical layer thickness-velocity curves were additionally plotted regarding the film thickness. In this study, also for the purposes of systematizing the results, the theoretical Reynolds equations were calculated on the basis of a full two-dimensional solution of finite differences for those surfaces covered with a lubricating film. The plotted theoretical curve read in the absence of slippage was equivalent to the classical Reynolds equation (Equation (1)). The theoretical results for full-slip conditions were calculated using the extended Reynolds equation model, with boundary conditions corresponding to a full slip, as in the case of Equation (2) [42]. Equation (2) uses the critical stress slip model. In this model, stresses of this type are assumed to be at zero. Comparing the terms of the equation on the right side of both equations, the one with the boundary conditions for a full slip (Equation (2)) was found to be only half that when using the boundary conditions without slip. As a consequence of these findings and assumptions, it can be assumed that the theoretical thickness of the layer under the boundary conditions of a full slip is smaller. The results of this finding are shown in Figure 1.


(1)
$$\frac{\partial }{\partial x}\left(h^{3}\frac{\partial p}{\partial x}\right)+\frac{\partial }{\partial y}\left(h^{3}\frac{\partial p}{\partial y}\right)=6u\eta \frac{dh}{dx}$$



(2)
$$\frac{\partial }{\partial x}\left(h^{3}\frac{\partial p}{\partial x}\right)+\frac{\partial }{\partial y}\left(h^{3}\frac{\partial p}{\partial y}\right)=3u\eta \frac{dh}{dx}$$


Based on Equations (1) and (2), it was assumed that two different lubrication scenarios were tested: “no-slip” lubrication and “full-slip” lubrication. Both scenarios were theoretically modeled using different versions of the Reynolds equation. From these results, it follows that the boundary conditions of full sliding will lead to a thinner lubricating film, compared to that in boundary conditions without sliding. This means that in a full-sliding scenario, the lubricating liquid is more widely “distributed” in terms of contact, leading to a lower lubricating film thickness. This may have important implications for a variety of applications where lubrication is critical. Therefore, both lubrication regimes are worth comparing.

The sliders of the device made of X90CrMoV18 steel and AlN/CrN/…/AlN/CrN provided a high layer thickness, which coincided with the classic theory of non-slip hydrodynamic lubrication. The highest layer-thickness results were also obtained for the AlTiN/CrN/Cr/…/CrN/Cr coating. This is an innovative and highly complex multi-layer coating created by the author in a combination that is based on theoretical assumptions, in order to obtain specific wetting properties. The thickness of the lubricating film created by the SiO_2_ and Cr sliders was almost the same and was slightly smaller than that of the AlTiN/CrN/Cr/…/CrN/Cr, AlN/CrN/…/AlN/CrN, and steel X90CrMoV18 sliders. The thickness of the lubricating film created by the CrN/AlN/…/CrN/AlN-coated slider was the lowest. In this case, its change according to speed is perfectly correlated with the theory of hydrodynamic lubrication for full-slip conditions. This shows that the molecular bonds between CrN/AlN/…/CrN/AlN and the lubricant are weak. The difference in the test conditions was due to the surface roughness. Other parameters for all the test trials, such as slider inclination, and the liquid parameters are the same.

The surface roughness of the CrN/AlN/…/CrN/AlN slider is higher than that for the other coatings (Table 1). Nevertheless, the roughness of this coating is almost an order of magnitude less than the measured minimum layer thickness. Slightly lower roughness was obtained for the X90CrMoV18 steel and the AlTiN/CrN/Cr/…/CrN/Cr and AlN/CrN/…/AlN/CrN coatings. The Cr and SiO_2_ coatings had the lowest roughness. Nevertheless, this parameter did not significantly affect the thickness of the oil film in relation to the coatings of X90CrMoV18 steel, AlTiN/CrN/Cr/…/CrN/Cr, and AlN/CrN/…/AlN/CrN.

Assuming these surface properties, it can be said that the small thickness of the lubricating film layer obtained with the CrN/AlN/…/CrN/AlN slider cannot result from high surface roughness. Otherwise, a higher roughness of the applied coating would increase the hydrodynamic effect, leading to an increase in film thickness. In this arrangement, the thickness of the lubricating film layer that is generated on the slider surface depends on interfacial and surface effects.

The relationship between the thickness of the lubricating film and the contact angle (CA) is shown in Figure 3. According to the accompanying figures, it can be concluded that the film thickness decreases significantly with the increase in the contact angle. In the case of the applied coatings for the test samples, it can be stated that they coincide with the theoretical calculations, except in the case of Cr and SiO_2_. For this coating, the contact angle is the second largest among all the adopted variants of sliders, but it produces a considerable thickness of the lubricating film.

Based on Figure 4, it can be assumed that the correlation between the thickness of the lubricating film layer and CAH is much better than in the case of CA. It performs much better in terms of the predictability of the lubricating film for individual materials. CAH shows a more stable and rectilinear nature in terms of the waveform. In the case of Figure 4, slight deviations of SiO_2_, Cr, and CrN/AlN/…/CrN/AlN from the approximation waveform can be noticed. An additional approximation of CA was performed using a sixth-order polynomial and, in the case of CAH, using a fourth-order polynomial.

Differences between the CrN/AlN/…/CrN/AlN sliders and the AlTiN/CrN/Cr/…/CrN/Cr sliders, which represent the two extreme abilities when forming a lubricating film, are shown in Figure 5. These were verified by carrying out tests with 99 wt % of glycerin. The introduction of glycerol from 65 to 99% by weight allowed for increasing the viscosity. The test was performed at 5 and 10 N. The slope was 1:1650. The results of two different loads are shown in Figure 5 and Figure 6 for the same coatings. The measurement results, including the AlTiN/CrN/Cr/…/CrN/Cr coating thickness of the slider, are well correlated, with the theoretical anti-slip curves in the specified speed range for two loads. However, the film thickness generated by the CrN/AlN/…/CrN/AlN slider is much smaller. Due to the large measurement errors of the interference images of the CrN/AlN/…/CrN/AlN slider at low speeds, Figure 4 and Figure 5 show only the thickness of the lubricating film layer, measured at higher speeds, starting from 5 mm/s. The minimum coating thickness shown in Figure 5 and Figure 6 is still about five times greater than the roughness of the CrN/AlN/…/CrN/AlN slider that is given in Table 1. It is, therefore, assumed that there is no direct contact between the two cooperating surfaces of the selected kinematic pairs.

A significant reduction in the thickness of the lubricating film for the CrN/AlN/…/CrN/AlN coating on the slider surface can be attributed to the hydrophobic properties of this coating and the intensification of the repulsion of lubricant particles at low feed speeds. Hysteresis of the surface contact angle of CrN/AlN/…/CrN/AlN and AlTiN/CrN/Cr/…/CrN/Cr with 99% by weight of glycerol is 35.2° and 45.2°, respectively, as shown in Table 3. It is assumed that the higher the value of the CAH parameter, the greater the thickness of the lubricating film. Figure 5 and Figure 6 show an interesting phenomenon wherein the resultant thickness of the lubricating layer obtained by the slider with the CrN/AlN/…/CrN/AlN coating is smaller than the theoretical full-slip curves. Therefore, the test was repeated using the Olalphaolefin oil. This oil has approximately the same dynamic viscosity value as 99% by weight of glycerol. However, these substances have different polarity values. Olalphaolefin oil is a non-polar oil, while glycerin is polar. This will extend the range for more comparison and will allow the researcher to assess the possibility of influencing the thickness of the oil film in the operating conditions of a piston oil engine.

Figure 7 and Figure 8 show the change in the thickness of the lubricating film layer for oliphaolefin oil in relation to the speed of movement at an inclination (slope: 1:1820) for loads of 5 and 10 N. In this scenario, all the accepted materials were used. Sliders with the material steel X90CrMoV18, AlTiN/CrN/Cr/…CrN/Cr, AlN/CrN/…/AlN/CrN, and CrN/AlN/…/CrN/AlN were tested under the same measurement conditions. With these coatings, there is a large variation in the contact angle (CA) of 26.1°; 12.4°, 20.6°, and 54.6°, respectively. In the case of CAH, these values are 31.2°, 28.2°, 30.9°, and 18.7°, respectively. The first three shells have divergent CA values but have very similar CAH values. Therefore, it is necessary to carefully analyze which of the parameters most closely reflects the theoretical prediction of CA or CAH thickness. The courses of the film thickness distribution for CAH correspond well to the classical theory of non-slip hydrodynamic lubrication. The difference in the contact angles between the materials of CrN/AlN/…/CrN/AlN and AlTiN/CrN/Cr/…/CrN/Cr is CA = 35.2°. In the case of sliders made of the X90CrMoV18 steel and AlN/CrN/…/AlN/CrN, the difference is CA = 5.5°. The Cr coating has a CA of 56.2° and a CAH of 25.1°. The SiO_2_ coating has a CA that is lower by 7.9° than that of the CrN/AlN/…/CrN/AlN coating. 

The thickness of the lubricating film layer for the oil, produced by selected sliders at different speeds, is shown in Figure 9 and Figure 10. The presented results prove that the CAH more closely reflects the relationship of the kinematic node regarding the effect of hydrodynamic lubrication produced by various sliding surfaces. This proves that CAH allows the better establishment of hydrodynamic lubrication conditions between two surfaces. In this arrangement, the CA is worse. In assessing the credibility of introducing the results of experimental research into mathematical models related to hydrodynamic lubrication, when assessing hydrophobic and hydrophilic properties, this is very important. Therefore, CAH gives a greater chance of credibility regarding changes in the oil film thickness in the elements of the cooperating kinematic pairs of the engine since the CA shows greater deviations from the values of theoretical models and approximation functions.

The test results for a sliding block covered with X90CrMoV18 steel, AlTiN/CrN/Cr/…CrN/Cr, and AlN/CrN/…/AlN/CrN correlate very well with the theoretical calculations for a pressing force of 1 and 2 N. The values for the film thickness in relation to the speed are characterized by a significant dispersion of the values and non-uniformity of the course. This coating has hydrophobic properties, while three of the coatings have hydrophilic properties. Between them, there are the oil film thickness distributions for Cr and SiO_2_. However, they have a more even distribution of film thickness values relative to the speed of the slider. The difference in the thickness of these layers results from the use of different surfaces. In this case, it corresponds well with their CAH, but not with the contact angle (CA). In the author’s opinion, this is the answer to the question of which parameter is more important for improving the model of the hydrodynamic theory of lubrication. In addition, the results shown in Figure 5, Figure 6, Figure 9 and Figure 10 indicate an unexplained phenomenon wherein the thickness of the film produced by the CrN/AlN/.../CrN/AlN slider with 99 wt % of glycerol or oil is much lower than in the theoretical full-slip values. It is difficult to explain this phenomenon of the stochastic course of values for hydrophobic coatings.

In [42], a detailed analysis of the influence of the non-wetted, stationary surface of a sliding object on its hydrodynamic properties was carried out using a model of critical shear stresses. In this work, the conditions of boundary slip were defined: full slip (τc = 0), one-way slip (a finite-valued constant τc, applied to the entire static surface of the slider), and directional slip (slip or no slip and slip direction depended on the local pressure gradient at the boundary). The tests were carried out on the basis of one-dimensional flow. Studies of this type on two-dimensional flows are presented in [43]. They describe the changes in the dimensionless load capacity W*, where ($Wh_{2}^{0}$/(Uη*B*^2^L) with respect to the dimensionless critical value of the shear stress τc* (h_0_ τc/Uη) for different inclinations of the rubbing pairs.

Large critical shear stresses of τc* > 1 correspond to non-slip conditions and the resistance curves approximate the Reynolds load values. The size of the ability to transfer the hydrodynamic load, i.e., to create a lubricating film, decreases when the limit slip begins on the stationary surface of a slider that is covered with any material. The disappearance of the hydrodynamic effect in the initial phase proceeds very quickly with an increasing level of slip, which is indicated by the decrease in τc*. Soon, this phenomenon is reduced to a minimum and the lubricating film-forming capacity increases again to further reduce τc*. Such a description is particularly important in the case of correcting the slip values for the various materials used for the elements of kinematic pairs of internal combustion engines. This is the basic condition for the planned shaping of the oil film, in terms of surface coverage and the thickness of the lubricating film that is obtained.

Figure 2 shows that the thickness of the lubricating film layer generated by the CrN/AlN/.../CrN/AlN slider with 65 wt % of glycerol solution coincides with the theoretical full-slip curve (i.e., τc = 0). This means that the critical shear stress of an EGC slider surface lubricated with 65 wt % values lead to a significant reduction in the hydrodynamic effect, i.e., load support, as shown in Figure 10. As a result, the thickness of the lubricating film layer formed by the CrN/AlN/.../CrN/AlN slider with 99 wt % glycerol or oil is much less. Under such conditions, the direction of lubricant slip in the pre-surface area of the slider is directed towards the inlet, which results in reduced grease pick-up and leads to a lower thickness of the lubricant film than under full-slip conditions. The two glycerol samples have similar chemical properties but show a large difference in dynamic viscosity. In this case, 99% by weight of glycerol has a higher viscosity. Greater critical shear stress 99% weights of glycerol on the surface layer of the CrN/AlN/.../CrN/AlN slider material can be attributed to its higher dynamic viscosity when using 65% by weight of glycerol.

The results of the experiment for selected sliders with different surface materials show the influence of the surface properties of various materials on hydrodynamic lubrication, in terms of CA and CAH. The ability to achieve hydrodynamic formation by a lubricating film is related to the adhesive force between the liquid and the solid surface. It takes place under conditions of fluid friction in the separation of cylinder walls, piston rings, and the piston. Mixed friction conditions are not considered in this paper. Grease particles can slide on the assumed surfaces only after overcoming the energy barrier. These parameters are defined by the CAO and CA. The magnitude of the energy barrier depends on the interfacial properties of the coating and the lubricant.

In [44], the energy barrier equation was derived, based on the basic principles of thermodynamics. It is expressed using CA and CAH:
(3)$$E=\frac{\gamma R}{2^{\frac{7}{3}}}(CAH{)}^{2}f\left(\theta \right),$$where:
(4)$$f\left(\theta \right)=\frac{(1+cos\theta {)}^{2}}{(1-cos\theta {)}^{1/6}(2+cos\theta {)}^{4/3}}$$

Calculation of the energy barrier, *E*, can be made using the parameters *θ* and *CAH*. However, the value of this parameter does not change significantly for the range from 20° to 140°. The parameter *f*(*θ*) practically remains unchanged within this range. To a large extent, the value of the energy barrier is due to the CAH. In the case of materials used in the construction of the kinematic elements of internal combustion engines, it is assumed that a contact angle greater than 90° means that the surface of the material is hydrophobic. Therefore, for the operating conditions of combustion engines wherein the maximum piston speeds achieved are temporarily much higher than in the test, the hydrophobic properties of the coatings do not necessarily have to be worse in terms of the formation of a lubricating film than for the hydrophilic coatings.

Based on Equation (1), it has been established that the smaller the CAH, the lower the barrier value. The main reason for this condition is the molecular interaction between the lubricating liquid and the surface layer of the coating. Limit slip occurs only above a CA of 140°. Slip length increases also occur above this value. A number of works on the methodology of research to assess the validity of the use of CA and CAH are given in [45,46,47,48,49,50,51]. According to the author, the research methodology adopted and the results obtained prove the significant potential of the theory of material selection, in terms of CA and CAH in its industrial use in internal combustion engines, in order not only to obtain the durability of these assemblies and low wear but also to significantly reduce the hydrodynamic friction losses occurring between the main components of internal combustion piston engines.

In internal combustion piston engines, the oil film is shaped depending on many engine operating parameters, the macro- and micro-geometry of assemblies, and the properties of liquids and materials. Currently, the latest trend is to assess the possibility of designing the thickness of the oil film and covering the sliding surfaces of the rings and the sliding surfaces of pistons and cylinders with an oil film, using the phenomena of hydrophilic and hydrophobic coatings. The influence of these parameters on the shape of the oil film and the wettability of the surface of kinematic materials is difficult to assess in dynamic conditions. There are two interfacial parameters, contact angle and contact angle hysteresis, which are effective for assessing these phenomena. In the tests, exemplary slider inclinations were adopted, reflecting the exemplary angles of inclination of the upper sealing ring in relation to the cylinder surface. The pressure force was selected due to the different thermodynamic conditions of engine operation and mainly due to the different pressures resulting from the movement of the piston rings on the cylinder wall. These data show the potential and the need to study these parameters in order to further reduce piston friction losses. Unfortunately, these tests are very complicated and are difficult to implement in a real combustion engine, but they can be carried out using simulation devices and specialized devices for measuring the contact angle and hysteresis of the contact angle.

## 4. Conclusions

Several coating materials with varying degrees of hydrophilic and hydrophobic features and with different CA and CAH parameters, along with three liquids, both polar and non-polar, were used in the tests, which provided contact angles from 12° to 75°. In the evaluation, in order to predict the thickness of the oil film, it was found that the key parameter is the hysteresis of the contact angle, which most closely reflects the phenomena of oil film formation than the contact angle in terms of the hydrophilic and hydrophobic characteristics. The results of these studies also reflect the basic theory of thermodynamics. It was found that the critical shear stresses constituting the limit slip barrier depend on the dynamic viscosity of the lubricant. The results of this research will allow future researchers to delve deeper into the hydrophilic and hydrophobic properties of coatings in oil film formation and design, which have not as yet been considered. This is very important, due to the need to further reduce the mechanical friction losses of the engine while maintaining the durability of kinematic assemblies. 

Based on the conducted research, it is concluded that the contact angle hysteresis (CAH) may be a more useful parameter in future studies of the wettability of coatings with liquids in internal combustion engines for several reasons: 

1. Information on surface heterogeneity: CAH provides information not only on wettability but also on surface heterogeneity. The surfaces of multilayer coatings are heterogeneous, which affects the quality of the obtained results. CAH provides valuable information that can help researchers to understand how the different surface properties of coatings respond to contact with lubricants of different chemical compositions. 

2. Measurement stability: CAH is more stable and is less prone to measurement errors, compared to a single CA measurement. This requires the measurement of both the upward and downward angles, which increases the reliability of the results. 

3. Dynamic behavior: CAH is more closely related to the dynamic behavior of the liquid on the surface, such as runoff, spreading, and evaporation (conditions that are more like those in engine mechanisms). This can be especially important in applications that require an understanding of these processes.

4. More information about liquid–surface interactions: CAH can provide more information about liquid–surface interactions such as adhesion, liquid fractionation, and liquid transport. 

Despite these advantages, it is worth remembering that CAH measurement is more complex and time-consuming, compared to CA measurement. Nevertheless, in many applications, especially those that require a deeper understanding of liquid–surface interactions, CAH can provide more valuable information.

## Figures and Tables

**Figure 1 materials-16-04092-f001:**
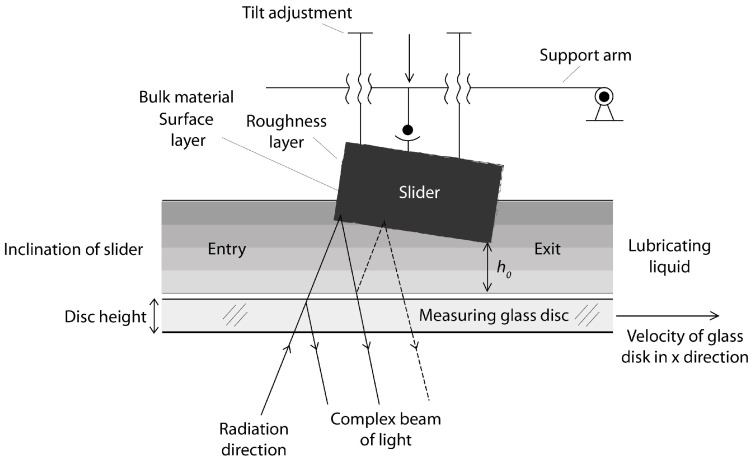
Overview view of the oil film testing station.

**Figure 2 materials-16-04092-f002:**
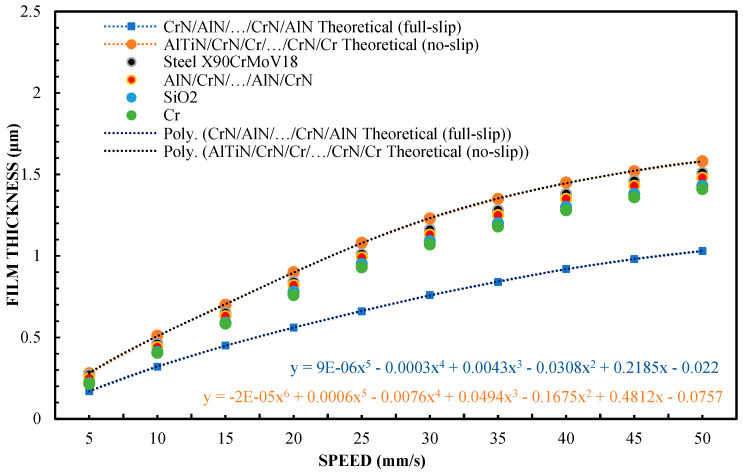
The thickness of the lubricating film layer, depending on the speed of movement for 65% by weight of glycerol and a load of W = 5 N.

**Figure 3 materials-16-04092-f003:**
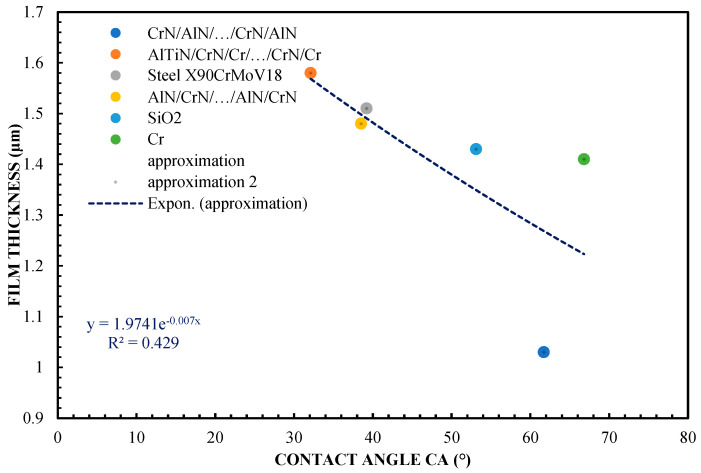
The correlation of layer thickness and contact angle (film thickness for the highest test values, in accordance with Figure 1)—65% glycerol.

**Figure 4 materials-16-04092-f004:**
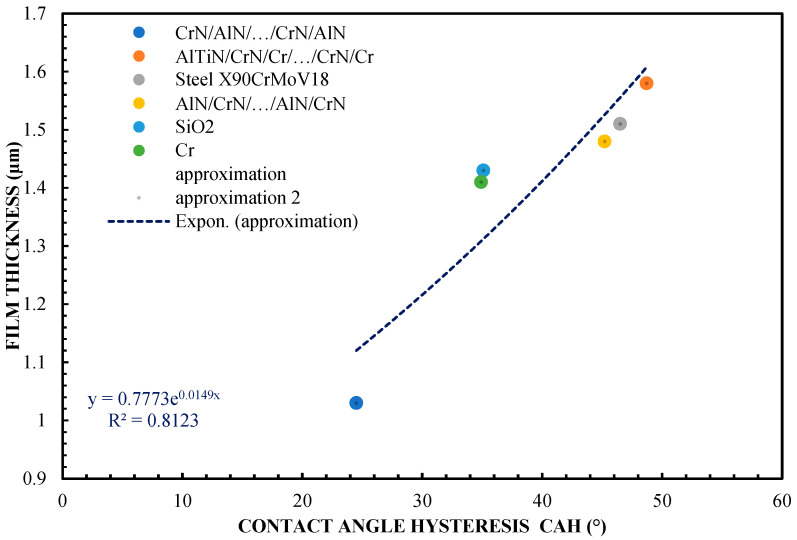
Correlation of the layer thickness and contact angle hysteresis—65% glycerol.

**Figure 5 materials-16-04092-f005:**
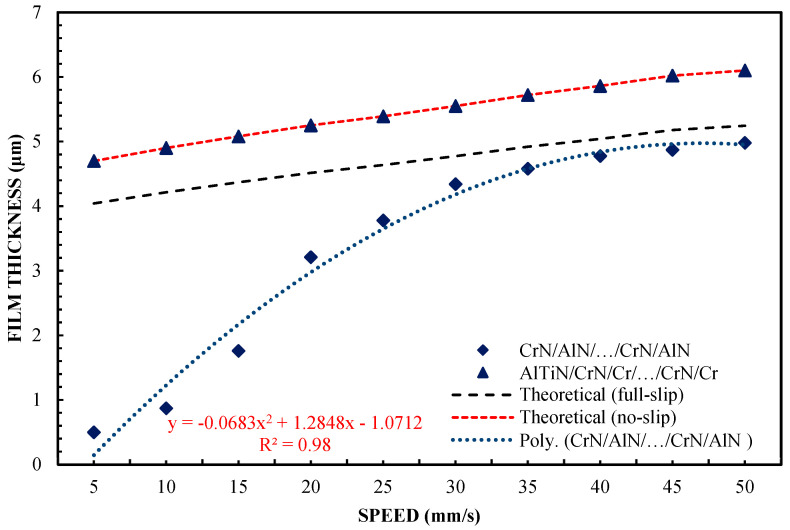
The thickness of the lubricating layer as a function of speed for 99% glycerin, with a load of W = 5 N.

**Figure 6 materials-16-04092-f006:**
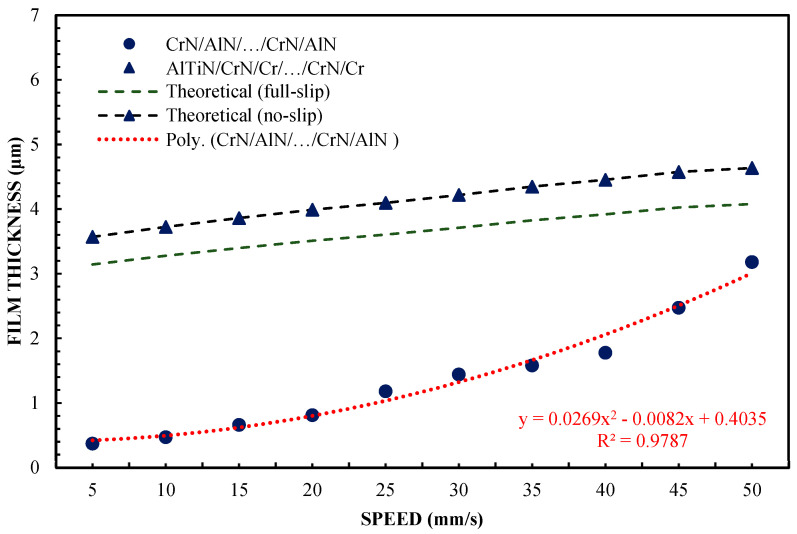
Thickness of the lubricating layer as a function of speed for 99% glycerin, with a load of W = 10 N.

**Figure 7 materials-16-04092-f007:**
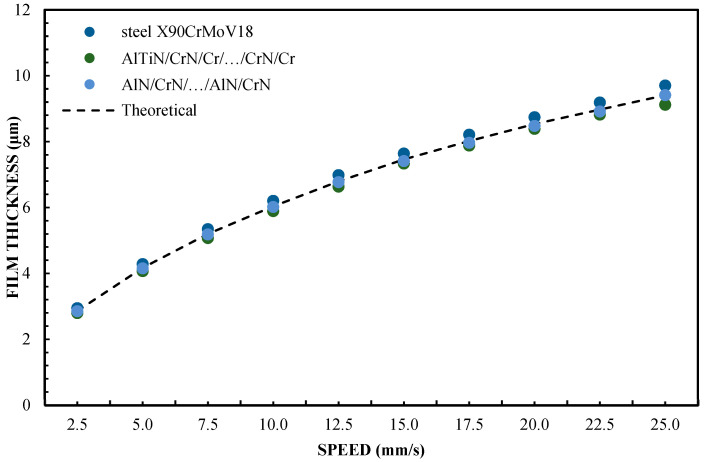
Lubrication film thickness for oil vs. speed for a 5 N load.

**Figure 8 materials-16-04092-f008:**
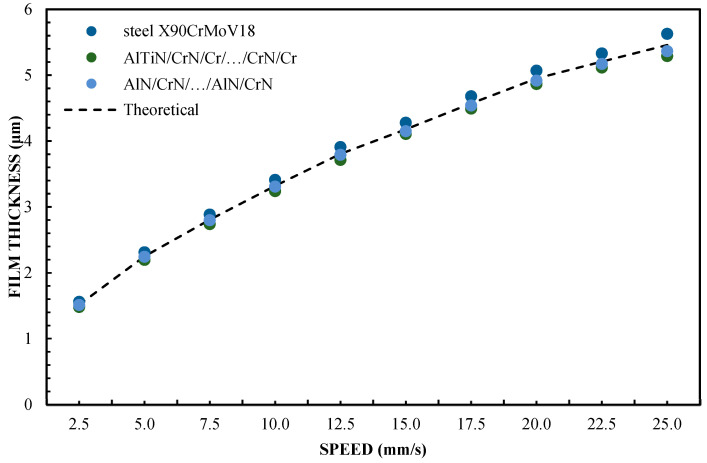
Lubrication film thickness for oil vs. speed for a 10 N load.

**Figure 9 materials-16-04092-f009:**
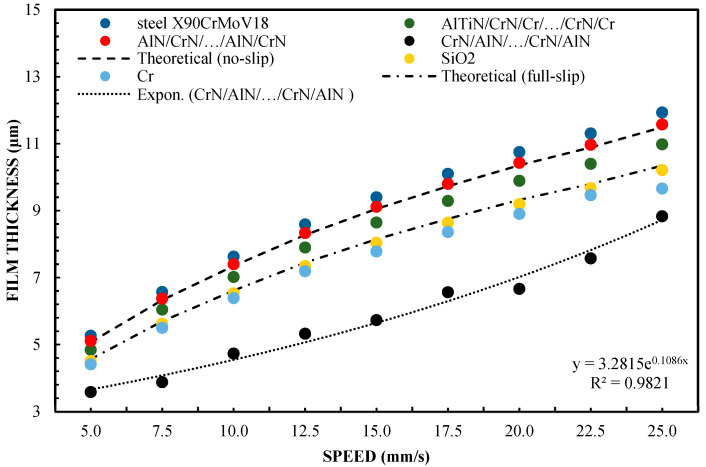
Lubrication film thickness for oil vs. speed for a 2 N load.

**Figure 10 materials-16-04092-f010:**
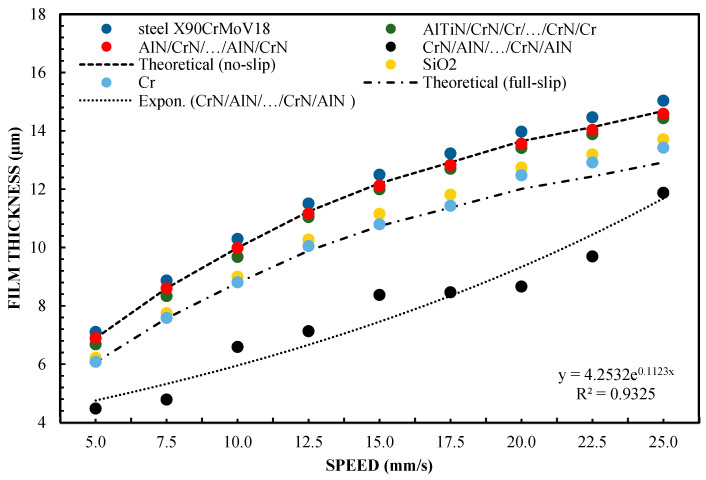
Lubrication film thickness for oil vs. speed for a 1 N load.

**Table 1 materials-16-04092-t001:** Properties of the base materials and counter samples.

Slider Type	Surface Layer	Bulk Material	Roughness (nm)	Number of Layers
1	Steel X90CrMoV18	Steel X90CrMoV18	123 ± 3	0
2	AlTiN/CrN/Cr/…CrN/Cr	Steel X90CrMoV18	121 ± 4	1—AlTiN, 5—CrN, 5—Cr
3	AlN/CrN/…/AlN/CrN	Steel X90CrMoV18	108 + 3	6—AlN, 5—CrN
4	CrN/AlN/…/CrN/AlN	Steel X90CrMoV18	139 ± 4	5—CrN, 5—AlN
5	Cr	Glass	4 ± 0.3	1—Cr
6	SiO_2_	Glass	5 ± 0.4	1—SiO_2_

**Table 2 materials-16-04092-t002:** Properties of the lubricants used in the tests.

Lubricant	Refractive Index	Dynamic Viscosity (22 °C, mPas)
Oil	1.46	840
99% Glycerol	1.47	704
65% Glycerol	1.45	14

**Table 3 materials-16-04092-t003:** The contact angle and contact angle hysteresis of the surface materials and lubricants.

Lubricating Liquids	Zipper Surface Material	Contact Angle CA (°)	Contact Angle Hysteresis CAH (°)
65% Glycerol	Steel X90CrMoV18	$39.2_{-5.4}^{+7.8}$	$46.5_{-3.2}^{+2.3}$
65% Glycerol	AlTiN/CrN/Cr/…/CrN/Cr	$32.1_{-5.8}^{+6.4}$	$48.7_{-2.1}^{+2.2}$
65% Glycerol	AlN/CrN/…/AlN/CrN	$38.5_{-4.6}^{+5.5}$	$45.2_{-2.4}^{+2.3}$
65% Glycerol	CrN/AlN/…/CrN/AlN	$61.7_{-4.8}^{+5.4}$	$24.5_{-1.4}^{+1.4}$
65% Glycerol	SiO_2_	$53.1_{-7.7}^{+8.4}$	$35.1_{-2.9}^{+2.6}$
65% Glycerol	Cr	$66.8_{-5.4}^{+5.1}$	$34.9_{-2.4}^{+1.7}$
99% Glycerol	Steel X90CrMoV18	$46.3_{-4.6}^{+4.8}$	$47.4_{-1.1}^{+1.1}$
99% Glycerol	AlTiN/CrN/Cr/…CrN/Cr	$42.7_{-3.2}^{+4.1}$	$45.2_{-1.3}^{+1.5}$
99% Glycerol	AlN/CrN/…/AlN/CrN	$45.2_{-3.7}^{+2.4}$	$47.4_{-1.0}^{+1.4}$
99% Glycerol	CrN/AlN/…/CrN/AlN	$69.8_{-3.4}^{+2.8}$	$35.2_{-1.1}^{+1.2}$
99% Glycerol	SiO_2_	$62.3_{-4.3}^{+3.8}$	$33.8_{-1.3}^{+1.6}$
99% Glycerol	Cr	$74.6_{-3.6}^{+4.2}$	$47.4_{-1.9}^{+1.1}$
Oil	Steel X90CrMoV18	$26.1_{-5.9}^{+5.8}$	$31.2_{-1.8}^{+1.6}$
Oil	AlTiN/CrN/Cr/…CrN/Cr	$12.4_{-2.2}^{+3.4}$	$28.8_{-1.4}^{+1.2}$
Oil	AlN/CrN/…/AlN/CrN	$20.6_{-1.5}^{+1.8}$	$30.9_{-1.2}^{+1.7}$
Oil	CrN/AlN/…/CrN/AlN	$54.6_{-1.6}^{+2.1}$	$18.7_{-1.9}^{+1.6}$
Oil	SiO_2_	$46.7_{-1.1}^{+2.1}$	$26.6_{-1.1}^{+1.3}$
Oil	Cr	$56.2_{-2.7}^{+1.5}$	$25.1_{-1.2}^{+1.2}$

## Data Availability

Not applicable.

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
