# Peer review of "The Theory of the Surface Wettability Angle in the Formation of an Oil Film in Internal Combustion Piston Engines"

_materials, 2023, doi:10.3390/ma16114092_

Round 1

Reviewer 1 Report

Needs to be improved.

Author Response

Reviewer Response 1. I've made all the suggested edits. The contact angle was measured with a goniometer. Simplified assumptions are given for the operation of a given device. They were not given for the engine because they are mathematically very extensive. The tests analyzed very thin layers of oil, such as exist in a real engine during the combustion of the fuel-air mixture. This corresponds to about 360-380 revolutions of the crankshaft between the most heavily loaded piston ring and the cylinder wall. Then the thickness of the oil film may be less than 0.5 micrometers. The inclinations were chosen randomly, similarly to the movement of the ring in the piston groove. The working conditions of the engine were simulated. Of course, it is impossible to determine them precisely, but you can only test sample unit pressures and film thicknesses that may occur in given engine operating situations. I entered this information into the text. Your critical look at the text allowed me to improve its quality and correct areas that needed correction. Considering your comments, I have made the suggested corrections and I am convinced that these changes have significantly increased the value and readability of my article. Your input is invaluable to me and I greatly appreciate your experience and professionalism. Thank you again for your valuable help and perceptiveness.

Reviewer Response 2. Dear Reviewer, I've added a drawing. I moved part of the conclusion to the discussion of the results. I corrected the drawings. I would like to thank you from the bottom of my heart for your pertinent comments and constructive suggestions regarding my article. Your critical look at the text allowed me to improve its quality and correct areas that needed correction. Considering your comments, I have made the suggested corrections and I am convinced that these changes have significantly increased the value and readability of my article. Your input is invaluable to me and I greatly appreciate your experience and professionalism. Thank you again for your valuable help and perceptiveness.

Reviewer 3. Dear Reviewer, I have made all your corrections. I also referred to the internal combustion engine and the connection with this research in the text. Thank you for your valuable advice. I edited the summary your advice was very valuable.

Reviewer 2 Report

The author of this manuscript studied the effect of wetting properties of several oil lubricants on their formation and properties in internal combustion piston engines. The obtained results of highlighted the need for studying these properties during the design stage of the piston engines. This work is original and well written and organized. I have the following three minor comments:

1. Can the author add a schematic drawing for the testing setup to section 2?

2.  In Figures 2-5: please correct the intercept and slope values in the line equations by changing "," to ".".

3. I believe the first 6 lines of the conclusion can be removed from this section. 

Author Response

(The authors gave the same response as above.)

Reviewer 3 Report

This study investigated the effect of wettability on the oil film and the results show that CAH is an important parameter to correlate the hydrodynamic lubrication. The experiments are complete. The results and discussion are in-depth. This is a good work. There are several problems though need to be raised.

1.      The theme of this study is about the internal combustion engine. Yet, the whole study setup shows little connection with ICE. Perhaps some important information is missing, like how the engine parts are installed in the apparatus, how the experiment mimics the real combustion cycle, or how the study setup relates to ICE, etc. If this study work isn’t tightly tied with the actual ICE configuration or operating condition, then probably discussing ICE to bring up the study motivation might just be sufficient.

2.      Some paragraphs are difficult to comprehend: Lines 43—44, 60—62, 66—67, 81—83, 169—170, 175—176, 363—368, 375—378

3.      Typos: should “coating thickness” on line 277 be “film thickness”? should “and CAH” on line 284 be deleted?

4.      Ambiguity: “bodies” in “The parameters …works on…” on line 60; “For this coating, …” on line 287; “particles” in “Lubricant particles …barriers” on lines 431—433 and in “Its value… and coating” on lines 433—434.

5.      Lack of explanation: “approximation”, “approximation 2” and “Expon” in Fig. 2

6.      Mislabeling: Eq. (1) on line 439 and (2) on line 441 should be (3) and (4), respectively.

7.      The style of reference list must be consistent.

8.      Abstract can be shortened.

See the aforementioned comments.

Author Response

(The authors gave the same response as above.)
